# Toward Competitive Serverless Deep Learning

Stefan Petrescu
Leibniz University Hannover
petrescu@vss.uni-hannover.de

Diego Albo Martinez
diego.albo.martinez@gmail.com

Jan S. Rellermeyer
Leibniz University Hannover
rellermeyer@vss.uni-hannover.de

## ABSTRACT

Machine learning is becoming a key technology to make systems smarter and more powerful. Unfortunately, training large and capable ML models is resource-intensive and requires high operational skills. Serverless computing is an emerging paradigm for structuring applications to benefit from on-demand computing resources and achieve horizontal scalability while making resources easier to consume. As such, it is an ideal substrate for the resource-intensive and often ad-hoc task of training deep learning models and has a strong potential to democratize access to ML techniques. However, the design of serverless platforms makes deep learning training difficult to translate efficiently to this new world. Apart from the intrinsic communication overhead (serverless functions are stateless), serverless training is limited by the reduced access to GPUs, which is especially problematic for running deep learning workloads, known to be notoriously demanding. To address these limitations, we present KubeML, a purpose-built deep learning system for serverless computing. KubeML fully embraces GPU acceleration while reducing the inherent communication overhead of deep learning workloads to match the limited capabilities of the serverless paradigm. In our experiments, we are able to outperform TensorFlow for smaller local batches, reach a 3.98x faster time-to-accuracy in these cases, and maintain a 2.02x speedup for commonly benchmarked machine learning models like ResNet34.

## KEYWORDS

Serverless, GPU acceleration, Machine Learning

**ACM Reference Format:**
Stefan Petrescu, Diego Albo Martinez, and Jan S. Rellermeyer. 2023. Toward Competitive Serverless Deep Learning. In *4th International Workshop on Distributed Infrastructure for the Common Good (DICG '23), December 11–15, 2023, Bologna, Italy*. ACM, New York, NY, USA, 6 pages. https://doi.org/10.1145/3631310.3633489

## 1 INTRODUCTION

Deep learning (DL) has emerged as a disruptive force to traditional services and business logic. It has not only enabled exciting applications like large language models [8, 37] or self-driving cars [6], but has also expanded the boundaries of what we considered solvable through software — it enabled solving problems that were previously thought impossible [11]. This progress has been largely driven by advances in hardware technology, which enabled deep learning models to significantly increase their performance by leveraging the combined power of compute and data. However, as we continue to tackle increasingly complex problems, how long can we accommodate the necessary compute for training?

Deep learning training is getting increasingly more challenging due to ever-increasing model sizes [32] and the corresponding increase in the need for training data (e.g., 175 Billion parameters for GPT-3 [8]). While the usual approach to training deep learning models is to reserve a cluster of machines in the cloud, this requires accurate knowledge of the resources necessary to train well in advance, which is challenging to estimate and often leads to over-provisioning of resources [3]. Even worse, the resources that provide the most benefit to deep learning training are significantly overprovisioned, with datacenter GPUs averaging under 20% utilization [10].

As a way to democratize access to the resource-intensive task of training deep learning models, serverless computing promises a low burden of adoption and the inherent agility and scalability of the cloud, with tremendous potential to contribute to the *common good*. Not only can it achieve horizontal scalability [24], but it also promises to do so while reducing management overhead, improving utilization, and cost efficiency. As such, it has the potential to make the training of ML models more accessible but also much more sustainable. However, the design and the stateless nature of serverless platforms are difficult to translate to the world of distributed machine learning and leverage the inherent advantages of fine-grained on-demand cloud resource sharing.

Serverless deep learning training is intrinsically challenging, as it requires overcoming (1) the limited compute capabilities of serverless functions, (2) and the communication overhead intrinsic to training in distributed settings [15]. For the former, deep learning workloads require access to accelerators (GPUs), which is currently not supported in serverless environments. For the latter, deep learning workloads can run for hours or even days, in contrast to traditional serverless workloads, which are typically short-lived and have moderate resource requirements. Given the difficulty of these challenges, no effective solution has been proposed, and even though serverless is a highly appealing choice, few studies have even considered serverless deep learning training.

To address the aforementioned challenges, we create a purpose-built serverless machine learning system –KubeML– that embraces GPU acceleration while optimizing for the communication overhead intrinsic to deep learning serverless workloads. In our experiments, we are able to outperform TensorFlow [1], especially with smaller local batches while allowing for higher resource density. Specifically, KubeML reaches a 3.98x faster time-to-accuracy with small batch sizes, and maintains a 2.02x speedup between the top results of both platforms for commonly benchmarked machine learning models like ResNet34. By enabling hardware-accelerated serverless functions and mitigating the communication overhead through design and optimization, we show that serverless could become the go-to medium for training deep models in the cloud.

*DICG '23, December 11–15, 2023, Bologna, Italy*
2023. ACM ISBN 979-8-4007-0458-1/23/12.
https://doi.org/10.1145/3631310.3633489

**Table 1: Categorization of existing work based on (1) communication overhead, (2) GPU acceleration.**

| System | Communication | GPU acceleration |
|---|---|---|
| PyWren [17] | ✗ | ✗ |
| Numpywren [33] | ✗ | ✗ |
| Locus [31] | ✓ | ✗ |
| Cirrus [9] | ✓ | ✗ |
| Siren [38] | ✓ | ✗ |
| lambdaDNN [41] | ✓ | ✗ |
| LambdaML [15] | ✓ | ✗ |
| GPU ESCF[19] | ✗ | ✓ |
| OSCAR [29] | ✗ | ✓ |

## 2  RELATED WORK & MOTIVATION

In the field of serverless ML, the problem of inference and model serving has received the most attention so far [2, 4, 14], with model training solutions being proposed more recently. In contrast, few works have been proposed exploiting GPU resources in a serverless environment, with only a few efforts having looked into the problem.

Proposed work, showcased in Table 1, (1) take advantage of serverless functions as a way to parallelize data processing workflows, whereas others (2) include ML or exploit GPU resources. For the former (1), for example, PyWren [17] offers a high level API to run map-reduce jobs on top AWS Lambda, using S3 for data I/O and event coordination. Numpywren [33] extends PyWren with optimizations for linear algebra operations. Similarly, Locus [31] also extends PyWren and explores the compromises between using slow (S3) or fast storage (Redis) for intermediate outputs in terms of performance.

For the latter (2), Cirrus [9] is designed as an extension of Py-Wren [17] and provides support for end-to-end ML workflows; the authors design a custom-made parameter server for maintaining the reference model and use S3 [27] for datasets, but do not test deep learning. Alternatively, Siren [38] uses S3 for all storage and focuses more on precise resource management and cost optimization. Similarly, lambdaDNN [41] highlights function allocation optimization, improving on the aforementioned systems (here, the authors use ZeroMQ to exchange parameter updates). Most recently, LambdaML [15] studies more in depth the dichotomy of pure-FaaS vs hybrid deployments for serverless ML, and proposes an end-to-end ML framework also incorporating deep learning using PyTorch [30]. However, none of the aforementioned systems are able to leverage GPU acceleration, and Jiang et al. [15] show that a single GPU instance is much faster and cheaper than multiple CPU lambdas.

Kim et al. [19] propose a serverless platform that uses IronFunctions coupled with NVIDIA-Docker to expose GPU resources to containers. Alternatively, OSCAR [29] uses serverless functions to allow clients to use remote GPUs as if they were local GPUs. OSCAR uses Kubernetes, Open-FaaS as their serverless platform, and rCUDA to virtualize and expose GPUs. Nevertheless, none of these solutions consider deep learning or distributed machine learning.

## 3  GPU ACCELERATION AND COMMUNICATION

Based on the characteristics of serverless applications and the context of DL training, we identify the key characteristics for KubeML, and focus on performance and usability: (1) utilization of GPU resources and (2) efficient communication model.

To enable GPU access for functions, we implement KubeML to run on Kubernetes, using NVIDIA-Docker to support GPUs. KubeML can train on multi-GPU (single-node) clusters as well as on multiple nodes without requiring any changes to the code. Next, to achieve strong convergence guarantees with limited communication overhead, we use a synchronous algorithm which combines characteristics from Elastic Averaging SGD (EASGD) [42] (synchronizing after multiple batches) with convergence properties from synchronous methods. Functions are deployed in multiple containers spread across machines in a cluster, having to communicate through the network. The characteristics of cloud networks [36] further increase the pressure to reduce communication but doing so without considering the model properties can negatively affect the convergence of the distributed training process [40].

### 3.1  Utilization of GPU resources

We implement KubeML on Kubernetes with multiple components to implement the distributed training as serverless functions. As shown in Figure 1, KubeML relies on different components to perform functions ranging from management operations to storage and tracking the training process. To enable straightforward interaction, all components expose a REST API, and communicate with each other using HTTP and JSON.

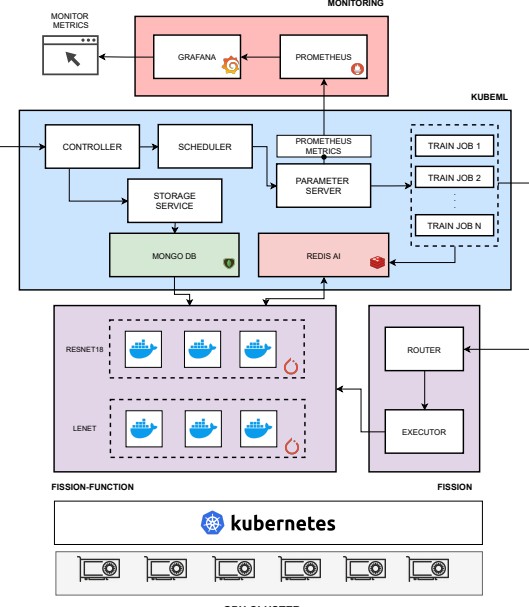

**Figure 1: Structure of the KubeML implementation; the platform is divided into different namespaces based on functionality.**

KubeML uses a data-parallel approach to accelerate training, which means that a replica of the network is initialized with the same weights as the reference model in each iteration, and trained on a different subset of the data. The functions then synchronize the models again and continue the training process. Users can reuse the same code they would use to train the network locally and upload it to the Kubernetes cluster with the help of the CLI. All main cloud providers offer their own hosted Kubernetes platform, which effectively makes migrating to a different provider trivial, with the same code and deployments being entirely transferable between clouds, and configurable to run in on-premise clusters. Moreover, Kubernetes also allows for local testing of applications, with tools such as MicroK8s [5] or kind [39], developed to enable deploying applications into a test environment and test them locally. We choose Fission [21] as our serverless platform because of its extensibility and its focus on performance [28]. Fission runs atop Kubernetes and implements the serverless paradigm using Kubernetes primitives. PyTorch code is wrapped using our custom made Python library, which provides a simple interface allowing using the same local code. We take advantage of PyTorch only using as much GPU memory as needed for training to allow multiple serverless functions to be allocated in the same GPU. With small models, a large fraction of the GPU resources remain unused, so this approach will increase resource utilization as well as improve performance without additional hardware resources.

Serverless functions are translated into the minimum entity of Kubernetes clusters, Pods, which can be described as a self-contained group of containers, all sharing common storage and network resources. Within a Fission function we find two containers. The environment is the container in which the function code will be executed. Initially the environment is nothing else than a generic Docker container with dependencies installed and a REST API. When a function is invoked, the container is specialized to serve a particular function, and will be used for that function alone until it is idle and its resources can be returned. The fetcher is in charge of loading the appropriate user code when a function is invoked through a trigger and specializing the environment, which loads the code provided and executes the main function. Generally, multiple instances of these pods are deployed simultaneously to serve requests in parallel and in isolation for a specific function type.

## 3.2 Overcoming the communication overhead

The traditional serverless programming model is most advantageous for conveniently parallel applications, where functions can proceed without having mutual data dependencies that require communication and coordination between the functions, unlike map-reduce jobs [14]. While simple and elegant, the model imposes noticeable limitations when developing applications with more strict communication needs. Serverless functions are not directly addressable and cannot be in a server role for communication which in practice means that functions cannot communicate with each other without an intermediate system to exchange information and/or synchronize execution. Storage platforms like S3, key-value stores like Redis, or application-specific parameter servers have been used in the past to facilitate inter-function communication. However, accessing these external components must be done through the network, which introduces extra latency in the application.

In distributed deep learning, the need for communication arises specifically from the need to periodically synchronize the model weights across workers to replicate the sequential behavior of families of algorithms like Stochastic Gradient Descent (SGD). To achieve this, two popular ways of syncing the models are adding the gradients and applying the global updates or model averaging. In terms of strictness, we can differentiate algorithms which force all workers to wait for each other before proceeding, like bulk-synchronous (BSP) algorithms, and others that slightly relax (stale-synchronous) or completely relax worker synchronization (asynchronous parallel). Further, communication can be influenced by the synchronization frequency: traditionally, synchronous algorithms like sync-SGD (S-SGD), synchronize the workers after each iteration or batch, but there are other more flexible options such as Local SGD, which only synchronizes after $k$ iterations, or one-shot averaging, which reduce the communication overhead by diminishing the number of synchronization points.

Consequently, to dimish the communication overhead, we use Local SGD [25, 34, 43, 44], also known as K-AVG SGD or Parallel SGD. In *Local SGD*, each worker trains for multiple iterations before syncing with the reference model, with the number of iterations before syncing commonly referred to as $k$. This parameter $k$ represents a balance between exploration (each worker exploring a concrete region of the loss space) with big $k$ and exploitation (all workers exploring the same region of the loss space) with small $k$. For merging the models into a single reference model, instead of aggregating the gradients like in other solutions, we take the approach of performing synchronous model averaging [20]. During the sync step, the models from all of the workers are averaged to obtain the new reference model, which effectively reduces the communication overhead by a factor of $k$ compared to synchronous SGD.

For exchanging model weights, a popular choice of topology is the parameter server architecture [13], in which one or multiple masters hold the weights of the model. These weights can optionally be sharded across multiple instances to increase throughput and for workers to pull and push their updated weights from them. Alternatives to the parameter server architecture include inherently more peer-to-peer topologies like AllReduce [16] in which workers are typically organized in a tree or ring configuration. None of the two options, however, are trivial to implement for a serverless setup because commercial platforms neither support creating collocated server images nor running functions that easily communicate with each other. KubeML is built around a parameter server architecture since in our setup we have control over the underlying Kubernetes cluster. Our server, however, also provides other critical services to the system, like centrally storing the training data in a MongoDB instance (given its performance on read operations) or using RediAI [7] as a high performance storage for the reference model and the function models during training. Further, our system ensures fault tolerance by continuously monitoring the state of the running functions to reprovision new functions in case of failures. Additionally, since KubeML needs to be able to host concurrent jobs from

different users, one key property of the platform is proper isolation. The server therefore creates the different jobs in new pods isolated from the rest of the system's components. Furthermore, it creates a separately managed pod that maintains the reference model for each submitted job. Instead of sharding parameters by name, it shards them by task, so each has its dedicated pod to update its parameters.

## 4 EVALUATION

To evaluate our system, we test it on a multi-GPU server. We test its performance during training tasks against TensorFlow, using small and medium sized networks on a variety of datasets. With these experiments we intend to study the viability of using Local SGD to train networks of different parameter sizes, and whether the increased parallelism of the training process can have a degrading effect on the convergence of the network.

### 4.1 Experimental Setup

**Platform**. For our multi-GPU server, we use a machine configuration with 2 NVIDIA RTX 2080 Ti and two 32-core AMD EPYC2 CPUs with SMT-2 enabled (128 hardware threads in total). As a system baseline, we compare the performance of our system against TensorFlow. We use the version 2.2 of TensorFlow and in all experiments use the MirroredStrategy to distribute the training load between both GPUs. The MirroredStrategy is an implementation of data parallel synchronous training, where the global batch is divided among the GPUs, which synchronize after each forward pass to keep a common view of parameters.

**Models & Datasets**. We use several networks representing common baselines and a variety of network sizes and architectures. We use the LeNet5 [23] as an example of a small network and train it using the MNIST dataset for written character recognition. To test the performance on bigger and deeper networks, we use the Resnet34 [12]. We use the CIFAR10 dataset [22], consisting of 50K train and 10K test images with 10 classes. Additionally, we train ResNet32 [12] to assess the performance for longer training tasks and the effect of the parameter $k$ on the convergence and the final results achieved.

**Metrics**. In our comparison against TensorFlow, our main metric is *Time-To-Accuracy* (TTA), defined as the time it takes for the validation accuracy to reach a certain amount. To settle on this amount for each of the networks, we train both until consistently reaching a plateau in terms of validation accuracy. We show these results in Figure 2, with the baselines for each network being: 99% for LeNet and 70% for the ResNet34. We also compare both systems by the final train and validation loss reached during training.

**Hyperparameters**. For the comparison with TensorFlow, we apply minimal transformations on the data and keep the hyperparameters fixed to avoid any possible difference in configuration. We compare both systems based on the per-worker or local batch. When training the LeNet, the learning rate is fixed at 0.01, while for the ResNet34 we fix it at 0.1. Both networks use a weight decay of 0.0001. As for dataset transformations, we standardize each datapoint feature-wise on both systems. For the optimizer, we use SGD with the learning rate and weight decay explained above, and a momentum of 0.9.

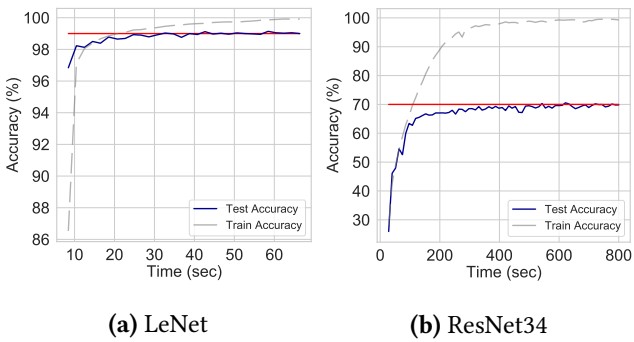

**(a)** LeNet                    **(b)** ResNet34

**Figure 2: TensorFlow convergence over time.**

### 4.2 Performance

Figure 3 showcases the results obtained when training on LeNet, where all batches reported correspond to the local batch size of each worker. For our system, we provide the best results achieved with the best parameter combination in terms of parallelism and $k$. All of the optimal parameter combinations for KubeML use a $k = \infty$, meaning that functions train locally for the entire duration of an epoch and only synchronize once before continuing the training process. We observe that the performance improvement has a tight relationship with the batch size: with bigger local batch sizes TensorFlow performs better than KubeML in terms of TTA. With smaller batches however, we see that KubeML outperforms TensorFlow and is 1.41x faster with a batch of 32, and 2.75x faster to the target accuracy with a local batch of 16. In these two cases, the best result is achieved with a parallelism of 8, that is, 4 models per GPU, showing that Local SGD is able to converge faster and without a loss in accuracy even with multiple workers scheduled per GPU.

Another relevant insight is the relationship between the train and the validation loss, also showcased in Figure 3. We see that even though our system consistently performs worse that TensorFlow in terms of train loss, it often results in a lower validation loss. This finding could corroborate the findings of Lin et al. [25], where the authors discuss that the local updates of Local SGD inject noise to the training dynamics, resulting in a convergence to flatter minima than traditional SGD. These flat minima are characterized by a better generalization than the sharp minima reached with other methods [18].

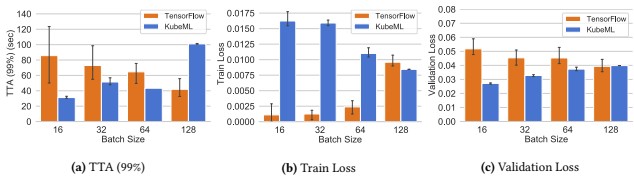

**(a)** TTA (99%)            **(b)** Train Loss            **(c)** Validation Loss

**Figure 3: Training performance comparison of LeNet.**

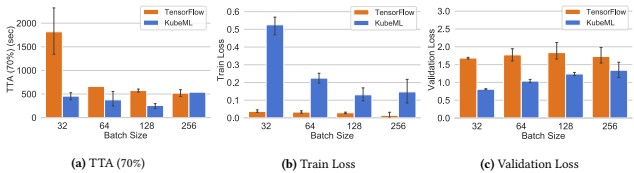

**Figure 4: Training Performance Comparison of ResNet34**

The benefits of using Local SGD should be accentuated when training bigger networks whose communication step takes a considerable amount of time when compared to the computing step, and this is the exact trend observed using ResNet34 as shown in Figure 4. With regards to the TTA, our system is consistently equal or better performing than TensorFlow, with the same improvement we saw with the LeNet taking place with small batches. With a local batch of 32, our system is 3.98x faster to 70%. Additionally, if we take into consideration only the best results from each system $b$ = 128 for KubeML and $b$ = 256 for TensorFlow), our system is still 2.02x faster to the target accuracy. Analyzing the losses in Figure 4, we reach the same conclusion as before. KubeML overfits less and generalizes better than TensorFlow.

**Resource Utilization**. Another important factor when training neural networks on GPUs is the utilization of resources. GPUs are expensive resources [26], and fitting one model per GPU often results in poor utilization, leading to a waste of both hardware and economic resources [20]. Our system makes multiple functions share the same GPU by means of the parallelism setting to improve resource usage. We evaluate how our system affects the usage of resources for the networks used previously.

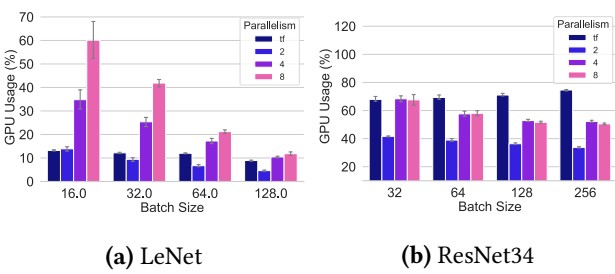

**Figure 5: GPU utilization comparison between TensorFlow and our system with different number of workers.**

We showcase our results in Figure 5, and observe that with more workers per GPU, the performance increases for LeNet, especially with smaller batches. This could be because interleaving small tasks on the GPU is more effective, reaching 6x better utilization than TensorFlow. However, with bigger networks, although observing that same improvement with small batches, KubeML stays on the same utilization level as TensorFlow. A cause for this could be the added communication time resulting from a bigger model (order of seconds for ResNet34 versus less than 10ms for LeNet) counteracting the throughput gains with more workers, since the addition of extra workers also adds extra communication overhead [35].

## 5 DISCUSSION & CONCLUSION

Deep learning training is getting increasingly more challenging due to ever-increasing model sizes and the corresponding increase in the need for training data. Tackling increasingly complex problems calls for more and more compute, as well as better ways to leverage it. Whether massive scale machine learning will become mission-critical enough for organizations to afford the best available hardware, or if it will fully tilt towards commoditization and the cloud remains to be seen. However, one thing is clear: for the majority of machine learning models, the cloud is and will remain to be an appealing solution due to its agility and scalability.

One solution to mitigate the resource-intensive nature of deep learning training is through the adoption of serverless computing — a paradigm which promises improved efficiency, no management overhead, and horizontal scalability. However, training deep learning models using serverless computing involves several challenges, as the high computational requirements of such workloads do not translate efficiently to this new world. To move toward a viable solution, we create a purpose-build platform –KubeML– that tackles these challenges.

KubeML extends the existing PyTorch programming model with support for serverless access to GPUs while accounting for the communication overhead between the functions typically encountered in the context of deep learning training. Our architecture is capable of handling common machine learning problems and outperforms TensorFlow on the same hardware in terms of training time and scalability. While the GPU scheduling is still a prototype and does not yet provide full virtualization of the accelerators, the design of KubeML is geared towards a high degree of multi-tenancy and thereby an increase of the utilization of these powerful but expensive resources. KubeML currently runs on Kubernetes and can therefore be easily deployed to cloud-hosted Kubernetes clusters. It does not run on commodity serverless platforms like Amazon Lambda, though. The most obvious reason is the current lack of GPU support.

A serverless cloud for distributed deep learning would also have to allow the collocation of serverless functions with centralized services (like our enhanced parameter server). The critical aspect of that is to have the ability to do so without sacrificing locality, which would further aggravate the issue of communication overhead. Additionally, capabilities like RDMA support can further reduce the communication latency for bulk transfers and make alternative architectures like AllReduce more competitive. While access to such capabilities and fine-grained control over them are currently not state-of-the-art and would require a rethinking of how to offer such capabilities without breaking the high-level abstraction of serverless functions, the growing interest in leveraging the serverles paradigm for deep learning makes it likely that more tailored solutions will enter the market in the near future.

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
