# OpenReview forum: "Toward Competitive Serverless Deep Learning"
_ACM.org/Middleware/Workshop/DICG — DICG 2023_

### Official Review · Reviewer_pscb · 2023-10-10
**The paper introduces SystemX, a serverless solution for data-parallel deep learning. The comparison with TensorFlow is not fully comprehensive because of differences in data-parallel training method implementations.**

**Rating:** 6
**Confidence:** 5

**Review:**

The paper introduces SystemX, an innovative serverless solution designed for data-parallel deep learning, aiming to address communication overhead and GPU utilization issues. In the study, the authors compare SystemX and TensorFlow's MirroredStrategy, which employs a data parallel synchronous training approach. The experimental results show that, in scenarios involving smaller batch sizes, SystemX achieves a faster time-to-accuracy and demonstrates better loss performance on the validation dataset compared to TensorFlow.
However, it is important to note that this comparison may not provide a fully comprehensive evaluation because SystemX employs local SGD, while TensorFlow's MirroredStrategy relies on synchronous training. To ensure a more equitable comparison, the authors should include an experiment involving the synchronous aggregation implementation of SystemX.
Moreover, it's worth mentioning that TensorFlow's MirroredStrategy utilizes an all-reduce implementation, while SystemX relies on the parameter server architecture. To better explain the advantages of SystemX, we suggest the authors consider using the same aggregation approach in both systems. This would help clarify whether the observed performance improvements are attributable to the serverless architecture of SystemX or the aggregation method employed.

---

### Official Review · Reviewer_ZVui · 2023-10-21
**Promising approach, requires refinement**

**Rating:** 6
**Confidence:** 5

**Review:**

The paper introduces SystemX, a platform designed to enhance deep learning training efficiency using a serverless computing paradigm.
Addressing the challenges posed by the computational demands of deep learning workloads, SystemX extends the PyTorch programming model to offer serverless access to GPUs while allegedly managing communication overhead. Through experiments on a multi-GPU server, SystemX demonstrated superior performance and scalability in comparison to TensorFlow, particularly benefiting from the Local SGD method where workers train on their data for an entire epoch and synchronize once. By running on Kubernetes, SystemX offers easy cloud deployment, aiming for better utilization of GPU resources.

Thanks for your submission to DICG. The paper is well written and tackles a nice topic, aligned to Middleware community and the Workshop.
I have however a few doubts about it. I would appreaciate if they were clarified in the camera-ready in case of acceptance.

- SystemX being an extesion of PyTorch, why do you compare to TensorFlow in the evaluation, and not PyTorch? If the reason is practical, i.e., because of TensorFlow `MirroredStrategy`, why did you not use TensorFlow in the SystemX Kubernetes Pods instead?
- How is the data partitioned across workers? DNN training is heavily dependent on statistical heterogeneity. If they are done differently in SystemX and TensorFlow, the comparison might not be fair.
- Did you really have to propose another framework? You mention that Kim et al. [19] proposes a serverless platform that exposes GPU resources to containers. Could you not just adapt it (or any other similar platform) to perform distributed ML?
- The claim about _overcoming communication overhead_ is somewhat misleading. Frequency of synchronization is a well known topic in distributed ML, which can affect the convergence speed depending on data distribution. While reducing synchronization frequency reduces communication, it is essential to be aware of the trade-offs involved.
- Saying you propose a "serverless" platform that depends on a Parameter *server* sounds like a contradiction.

Minor:
- Font size in Figures are very small and do not print well. Please, make them bigger.
- You mention _"models from all of the workers are averaged to obtain the new reference model, which effectively reduces the communication overhead by a factor of $k$ compared to synchronous SGD."_ Later on, you say that you adopt $k=\infty$. Strictly speaking, this would make the communication overhead go down to zero, which is clearly innacurate. Please, fix the statement.

Overall, the paper is well written and presents a nice proposal.
Evaluation though pertains mostly to having a distributed setup rather than a centralized one, and not necessarily to the "serverless" property. The serverless aspect seems only relevant to the practical deployment/implementation (multi-tenancy, Kubernetes + Fission).
In any case, this paper could incite nice discussions in the Workshop.